# Hydrothermal Crystallization of Bismuth Oxychlorides (BiOCl) Using Different Shape Control Reagents

**DOI:** 10.3390/ma14092261

**Published:** 2021-04-27

**Authors:** Enikő Bárdos, Viktória A. Márta, Szilvia Fodor, Endre-Zsolt Kedves, Klara Hernadi, Zsolt Pap

**Affiliations:** 1Department of Applied and Environmental Chemistry, University of Szeged, Rerrich Béla tér 1, HU-6720 Szeged, Hungary; bardosenci@gmail.com (E.B.); martaviktoria95@gmail.com (V.A.M.); fod_szilvia@hotmail.com (S.F.); 2Institute of Environmental Science and Technology, University of Szeged, Tisza Lajos krt. 103, HU-6720 Szeged, Hungary; 3Faculty of Physics, Babeș-Bolyai University, M. Kogălniceanu 1, RO-400084 Cluj-Napoca, Romania; kedves.endre91@gmail.com; 4Nanostructured Materials and Bio-Nano-Interfaces Center, Institute for Interdisciplinary Research on Bio-Nano-Sciences, Babeș-Bolyai University, Treboniu Laurian 42, RO-400271 Cluj-Napoca, Romania; 5Department of Biosystems Engineering, Faculty of Engineering, University of Szeged, Moszkvai krt. 9, H-6725 Szeged, Hungary; 6Institute of Physical Metallurgy, Metal Forming and Nanotechnology, University of Miskolc, 3515 Miskolc-Egyetemváros, Hungary

**Keywords:** bismuth oxychloride, photocatalytic activity, surfactants, surface tension

## Abstract

Bismuth oxychloride photocatalysts were obtained using solvothermal synthesis and different additives (CTAB—cetyltrimethylammonium bromide, CTAC—cetyltrimethylammonium chloride, PVP–polyvinylpyrrolidone, SDS–sodium dodecylsulphate, U—urea and TU—thiourea). The effect of the previously mentioned compounds was analyzed applying structural (primary crystallite size, crystal phase composition, etc.), morphological (particle geometry), optical (band gap energy) parameters, surface related properties (surface atoms’ oxidation states), and the resulted photocatalytic activity. A strong dependency was found between the surface tension of the synthesis solutions and the overall morpho-structural parameters. The main finding was that the characteristics of the semiconductors can be tuned by modifying the surface tension of the synthesis mixture. It was observed after the photocatalytic degradation, that the white semiconductor turned to grey. Furthermore, we attempted to explain the gray color of BiOCl catalysts after the photocatalytic decompositions by Raman and XPS studies.

## 1. Introduction

BiOX (X = Cl, Br, I, bismuth oxyhalogenide) catalysts, including BiOCl (bismuth oxychloride) constitute a new generation of semiconductors with high potential as photocatalysts. These materials have a tetragonal matlockite structure, where each bismuth atom is surrounded by four oxygen and four chloride atoms, forming [Cl-Bi-O-Bi-Cl] layers held together by Van der Waals forces [1]. Recently, BiOCl was investigated more intensively, because of its beneficial properties, such as, low cost, simple synthesis methods and adequate band gap energy, similar to that of TiO_2_ (3.2 eV) [2]. Several methods are available in the scientific literature to achieve photocatalytic activity under visible light irradiation for TiO_2_ and BiOCl, such as the induction of oxygen vacancies [3], noble metal deposition or shape-controlled synthesis [4,5,6,7]. BiOCl presents promising results in visible-light-driven photocatalysis [8,9,10] as well for degrading organic pollutants such as phenol [9], pharmaceuticals [11] or even dyes [12].

By applying a shape-controlled synthesis it is possible to favor the growth of different crystallographic planes, which provide beneficial properties in desired applications. The (001) (102) and (110) crystallographic planes of BiOCl materials were already investigated and it was found, that (102) showed lower value of the conduction band minimum/ higher valence band maximum, compared to (001). The latter one also has a smaller band gap, while the redox potential of the photoinduced hole is higher. It was found that (110) possessed a stronger internal electric field compared to (001), which could increase the lifetime of the charge carriers [5,13,14].

According to the scientific literature sodium dodecyl sulfate (SDS), polyvinylpyrrolidone (PVP) [15], and urea [16] were already used as shape-controlling agents during the hydrothermal crystallization of BiOCl materials. SDS, cetyltrimethylammonium chloride (CTAC) and PVP molecules are composed from a hydrophilic and a hydrophobic part. The molecule’s hydrophilic groups could be adsorbed on the surface of the crystallographic planes of the BiOCl. The adsorbed surfactants block the epitaxial growth of the exposed crystal facets thus the morphology and the final crystal size can be controlled as well. Moreover depending on the hydrophobic functional group the distance among the hydrophilic parts can be regulated which determines the nanosheets’ thickness [15,17].

Thiourea (TU) coordinates to bismuth resulting Bi^3+^-(TU)_n_ complexes, which can participate in the formation of hierarchical microstructures (e.g., spheres/flowers). The formation process of the BiOCl microspheres or microflowers consist of two steps: first, the complexation between Bi^3+^ ions and TU leads to the formation of Bi-TU complexes at room temperature in ethylene glycol (EG). During the complexation process the solution’s color turns to yellow from transparent when Bi(NO_3_)_3_ and TU solutions are mixed together. The second step is when the Bi-TU complexes decomposes during the solvothermal crystallization process to form the BiOCl crystals [18,19].

In our previous work, we proved that the surface tension of the synthesis solution could affect the specific surface area, crystallographic planes ratio, primary and hierarchical sizes and the photocatalytic activity of BiOBr [20]. So, it is mandatory to investigate the relationship between the surface tension and the previously mentioned properties.

As it was discussed above, different additives could lead to different properties in the case of BiOCl solvothermal crystallization, hence a comparison of the effects within the same synthesis system would lead to important general information concerning this interesting compound. Thus, in this study solvothermal synthesis was used to obtain BiOCl applying different shape controlling agents, more precisely CTAC, SDS, PVP, urea and thiourea, respectively. The main aim of the current study was to comparatively investigate the effect of different compounds on the surface tension, crystal structure, morphology, band gap energy, surface tension of the synthesis mixture and to investigate the photocatalytic activity of the as-obtained semiconductors towards phenol and RhB (Rhodamine B).

## 2. Materials and Methods

### 2.1. Synthesis

Bismuth oxychloride samples were synthesized by a solvothermal crystallization method. 2.78 g Bi(NO_3_)_3_·5 H_2_O (VWR International Ltd., Debrecen, Hungary, 98.0%) was dissolved in 50 mL ethylene glycol (VWR International Ltd., Debrecen, Hungary, 99.5%), afterwards a calculated amount of surfactant was added such as polyvinylpyrrolidone (PVP K30) (Molar Chemicals, Halásztelek, Hungary, 99%), sodium dodecyl sulfate (SDS) (Biolab Inc., Budapest, Hungary 98%), cetyltrimethylammonium chloride (CTAC) (Sigma Aldrich, Schnelldorf, Germany 98%), urea (U) (Molar Chemicals, Halásztelek, Hungary, 99%) and thiourea (TU) (Sigma Aldrich, Schnelldorf, Germany 99%). Then 0.42 g of KCl (VWR International Ltd., Debrecen, Hungary 99.0%) was added ensuring, that the Bi:X ratio was 1:1. Each solution was stirred for 15 min at 50 °C and then placed to a Teflon^®^ lined autoclave and put into an oven (BD 115, Binder, Thomas Scientific, Swedesboro, New Jersey, NJ, USA) at 120 °C for 3 h. When the crystallization process was over, the autoclave was cooled down to room temperature. The resulting materials were washed with 0.5 L ethanol (VWR, International Ltd., Debrecen, Hungary 96%) and after that with 0.5 L distilled water using a vacuum filtration system.

The samples were dried at 40 °C for 12 h. The nomenclature of the materials was derived as follows: *BiOCl_applied additive*.

### 2.2. Characterization Methods

#### 2.2.1. Diffuse Reflectance Spectroscopy (DRS) 

This technique was used to measure optical properties of the samples. The spectra of the samples were acquired applying a V650 spectrophotometer (JASCO, Wien, Austria) equipped with an integration sphere (ILV-724, λ = 250–800 nm). The band gap energy values of the semiconductors were estimated by the Kubelka-Munk approach and the Tauc plot [21].

#### 2.2.2. Scanning Electron Microscopy (SEM) 

A S-4700 Type II FE-SEM (Hitachi, Tokyo, Japan) equipped with a cold field emission source operating in the range of 5–15 kV was used to record micrographs and to investigate the morphology of the particles. The obtained materials were mounted on a conductive carbon tape, which was attached to an aluminum holder. We determined the hierarchical crystal size distribution using ImageJ software based on more SEM micrographs, while counting at least 100 particles, and from these data the particle size distribution histograms were created.

#### 2.2.3. Powder X-ray Diffractometry (XRD) 

XRD analysis was implemented on a Miniflex II diffractometer 2θ° = 20–60°, λ (CuK_α_) = 0.15418 nm) using 2(2θ°) min^−1^ scanning speed (Rigaku, Neu-Isenburg, Germany). The mean primary crystallite mean size values were estimated using the Scherrer equation [22].

#### 2.2.4. Specific Surface Area 

The catalyst parameter was determined by N_2_ adsorption at 77 K, using a BELCAT-A device (Microtrac MRB, Osaka, Japan). The specific surface area was calculated via the BET method.

#### 2.2.5. XPS Measurements 

These were carried out on a Phoibos 150 MCD system (SpecsBerlin, Berlin, Germany) using a monochromatic Al-Kα source (1486.6 eV) at 14 kV and 20 mA and the X-ray source with 200 W, while the pressure in the analyzer chamber was reduced below10^−9^ mbar. The investigated materials were placed on a double-sided carbon tape. The high resolution O1s and Bi4f spectra were obtained using analyzer pass energy of 20 eV in steps of 0.05 eV for analyzed samples.

#### 2.2.6. Raman Spectroscopy 

A Thermo Scientific (Waltham, MA, USA) DXR Raman microscope was used with a diode-pumped frequency-doubled Nd:YAG (10 mW) at 532.2 nm wavelength of the laser. The used confocal aperture was 50 μm pinhole. Recording the spectra autoexposure operating mode was used, within 100 signal/noise ratio. The spectra were taken by mapping method on 5 μm × 5 μm area.

#### 2.2.7. Surface Tension 

The surface tension of the synthesis solution was carried out by using a 3.5 mL stalagmometer (Wilmad Labglass, Vineland, NJ, USA) and Milli-Q water (Sigma Aldrich, Schnelldorf, Germany) as the reference. The density of the solutions was measured by pycnometer (10 mL) with a thermometer. The following Equation (1) was applied during the calculations:(1)$$\gamma =\frac{\gamma_{w\;}\cdot \rho \cdot n_{w}}{\rho_{w}\cdot n}$$
where γ, γ_w_—the surface tension values (Pa·s), n, n_w_—the number of counted liquid drops, ρ, ρ_w—_density of the liquid (g·cm^−3^) and w—water.

### 2.3. Evaluation of Photocatalytic Efficiency

A double-walled glass reactor with a thermostatic jacket (T = 25 °C) was used to measure the photocatalytic activity of the samples. The reactor was surrounded by 6 × 6 W black-light lamps (λ_max_ ≈ 365 nm) or 4 × 24 conventional compact fluorescence lamps (Düwi 25920/R7S-24W). 1 M sodium nitrite (Molar Chemicals, Halásztelek, Hungary min. 99.13%) aqueous solution was circulated during the visible light degradations to modify the emitted spectrum of the lamps. The NaNO_2_ solution absorbs the ultraviolet (UV) photons, therefore NaNO_2_ acts like a cut-off filter and provide just visible light irradiation (Appendix A). During the UV tests water was circulated in the thermostatic jacket. The suspension (C_0, phenol_ = 0.1 mM; C_0, RhB_ = 0.3 mM; C_suspension_ = 1.0 g∙L^−1^; total volume of the suspension V_suspension_ = 130 mL, degradation time = 2 h) was continuously stirred and purged with air to keep the dissolved oxygen concentration constant during the whole experiment. 1.5 mL sample was taken and centrifuged for 3 min at 13,400 rpm and filtered with 0.25 µm syringe filter (Filtratech, Sigma Aldrich, Schnelldorf, Germany). The concentration of RhB was followed using a JASCO V-650 spectrophotometer. The detection wavelength for RhB was fixed at 554 nm. The concentration of phenol was followed by High Performance Liquid Chromatography (HPLC, L-7100, Merck-Hitachi, Darmstadt, Germany) equipped with a low-pressure gradient pump, and a Merck-Hitachi L-4250 UV–Vis detector and a Lichrospher R_p_ 18 column, while the applied eluant was a mixture of methanol/water (50:50 v/v) and 210 nm was set as the detection wavelength.

## 3. Results and Discussion

### 3.1. Evaluation of Crystal Phase Composition, Primary Crystallite Size and Specific Surface Area

The crystal phase composition and the average primary crystallite size was determined by X-ray diffraction measurements. Figure 1 shows the XRD patterns of the prepared samples and the tetragonal structure of BiOCl (JCPDS card No. 82-0485), while no other relevant signals were noticed.

The XRD patterns of the obtained BiOCl_∅ sample—which was synthesized without additives—shows intense diffraction peaks of (101), (110) and (102). In contrast, when additives were used during the BiOCl synthesis, the intensity ratios of the (101), (110) and (102) crystallographic planes changed, specifically the reflections of (101) and (102) (Table 1). These changes may arise from morphological differences, which would impact the photocatalytic activity.

The primary crystallite size values were estimated as described in the experimental section and were listed in Table 1. The results disclose clearly that the surfactants presence had an impact on the BiOCl crystallization growth. Compared to the additive free sample in each case smaller particle size was obtained (e.g.,: BiOCl_∅ 21.1 nm, BiOCl_PVP 16.5 nm). It should be mentioned that besides the BiOCl_CTAC had the smallest primary crystallite size (11.9 nm).

Moreover, the connection between the crystallographic planes’ ratios and the primary crystallite size was also studied. The (102)/(110) the primary crystallite size varied along a minimum curve (Figure 1A). In case of (102)/(101) a similar trend was observed (Figure 1B,C). This could mean that a specific additive could stabilize the crystallite at a specific orientation, conferring the possibility to form differently shaped hierarchical crystals as they are aggregating/growing. Suggesting that there will be differences in band gap energy and photocatalytic activity.

### 3.2. Morphological Characterization

After the structural characterization, the morphology of the particles was investigated. All the materials showed hierarchical microcrystalline structure, which were composed from fine nanoplates. In the absence of any surfactants, the SEM micrographs of BiOCl showed microspherical morphology as it is shown in Figure 2. When CTAC or SDS were used, no significant morphological changes were observed, while the median hierarchical crystal size increased from 1.5 µm to 2.4 µm as shown in Table 1, while in the case of BiOCl_CTAC the size distribution became wider (from 1.00–3.00 µm to 1.00–4.50 µm). When PVP or TU was used to obtain BiOCl, the main morphology changed from microspheres to isomorphic “microflowers”. The median hierarchical crystal size was the smallest here (decreased from BiOCl_Ø 1.5 µm to 0.7 µm).

When TU was applied in the preparation of BiOCl, hierarchical microflower shape was obtained, similarly when PVP assisted BiOCl, even though using TU the microflowers were aggregated. The median hierarchical crystal size increased from 1.5 µm (BiOCl_Ø) to 2.0 µm, but in both cases the size distribution became narrower (from 1.00–3.00 µm to 0.50–1.50 and 0.50–2.50 µm). This could be advantageous for the photocatalytic activity, because these samples might possess higher specific surface area. Investigating the morphology of BiOCl_U, a few cube-like hierarchical structures appeared as well (Figure 2. red frames). This phenomenon was already noticed in our previous work, when urea was applied as additive during the hydrothermal crystallization of BiOBr [20].

As it can be seen the SEM micrographs the used additives influenced the particle morphology and also the orientation of the crystals as well. It is already known in the literature that each of the additives may impact several important features, such as:Oriented growth of crystals by blocking a crystallographic direction through surface anchoring.Aggregation state of the primary crystallites, the possibility of Ostwald ripeningHierarchical particle size and specific surface area tuning, by non-selective surface adsorption

The above listed parameters could contribute in band gap energy modulation as well. All the important issues listed above foreshadows that a unique correlation will be available between the different properties of the material and an additive-related parameter, such as the surface tension of the synthesis mixture, an aspect discussed later in the present work.

The specific surface area values are summarized in Table 1. As expected, the BiOCl_CTAC with higher hierarchical particle size had the lowest specific surface area—5 m^2^·g^−1^. In the case of BiOCl_SDS, a similar hierarchical particle size was measured, but its specific surface area approached that of BiOCl_Ø crystallized without additives (16.5 m^2^·g^−1^). Interestingly, the BiOCl_TU sample possessed a low specific surface area of 8 m^2^·g^−1^, but contained smaller hierarchical particles, meaning that the individual crystallites are very tightly arranged. The use of TU and PVP increased the specific surface area to 25 m^2^·g^−1^ for sample BiOCl_TU, while a 57 m^2^·g^−1^ value was registered for BiOCl_PVP.

### 3.3. Assessment of the Band Gap Energy

By analyzing the acquired data it was concluded that the used additives did not have significant effect on the band gap energy values (Figure 3), which were near to those available from the scientific literature (BiOCl: 3.2–3.5 eV) [23,24,25]. The band gap energy values deduced that the samples will be active under UV irradiation.

The connection between the crystallographic planes’ ratio values and the band gap energy was also investigated. As it can be seen in Figure 3B, when the (102)/(110) ratio increased, the band gap energy values increased as well. Similar trend was noticed, when the (101)/(110) planes’ ratio was investigated, so it was concluded, that (110) has higher band gap energy, compared to (101)—Figure 3C. When the (101)/(102) ratio increased the band gap energy decreased Figure 3D, pointing out that (102) possess lower band gap energy compared to (101).As it can be noticed in Figure 3B–D there is one point, which belong to the BiOCl_CTAC sample and this point did not fit into the trend.

### 3.4. Photocatalytic Efficiencies of the Obtained BiOX Materials

The effect of additives on the semiconductor’s photocatalytic properties was determined in the photocatalytic decomposition of both RhB and phenol. Figure 4 shows the photocatalytic conversion of the RhB and phenol under both irradiation types. It was observable that the degradation of RhB was more efficient under visible light than UV irradiation. Using additives during the synthesis caused an increase in the photocatalytic efficiency degrading phenol. 

In case of BiOCl_CTAC the reason of the low dye degradation was the low specific surface area/ high hierarchical particle size. The conversion of RhB under UV irradiation on sample BiOCl_SDS was near to that of BiOCl_Ø. The highest degradation efficiency values were achieved by BiOCl_TU and BiOCl_PVP samples (69% and 91%, respectively), but it is also important to mention that the adsorption on these semiconductors was also high (40–50%).

The reason of the high photocatalytic efficiency was the low hierarchical particle size and high specific surface area. In case of visible light degradation experiments (Figure 4B), all the catalysts decomposed the RhB, but BiOCl_TU and BiOCl_PVP proved to be the most effective, degrading the dye in 30 min under visible light. The high degradation efficiency in visible light could be explained by the well-known dye sensitization process [26,27]. It is important to mention, that intermediates were formed, when visible light irradiation was used to degrade RhB. Except for samples BiOCl_CTAC and BiOCl_U the UV-Vis spectra of RhB showed the complete degradation of RhB and of the possible intermediates as well.

RhB presents high adsorption yield towards BiOCl, therefore phenol was selected as contaminant, due to its poor adsorption properties on the surface of semiconductors. It can be seen in Figure 4C) that the adsorption was very low (1–2%). The overall degradation of phenol was lower compared to the removal of RhB, the highest activity was achieved under UV irradiation by the sample BiOCl_PVP (31%). However, under visible light the decomposition of phenol was very low, only 8% was achieved using the BiOCl_U and BiOCl_TU samples.

To understand better the role of the specific surface area, the surface-normalized degradation of the RhB and phenol under UV irradiation was also investigated. The results from surface-reported degradation efficiency of RhB and phenol pointed out that the quality (the available catalytic centers) of the surface changed with the additives. First, we discuss in detail the normalized surface values of the RhB degradation under UV irradiation. BiOCl_CTAC seems that the quality of the surface increased compared to the BiOCl_ Ø (with 16.00 m^2^·g^−1^, degraded 1.35 mM·m^−2^ RhB), as the samples BiOCl_CTAC with smaller specific surface area (5.00 m^2^·g^−1^) degraded more RhB/ surface unit (4.82 mM·m^−2^). BiOCl_SDS possessed similar specific surface area, such as the BiOCl_ Ø and achieved higher normalized activity values, when RhB was degraded (2.83 mM·m^−2^). Even though the sample BiOCl_PVP has the highest specific surface area value, presents the lowest normalized activity value. BiOCl_PVP had 57 m^2^·g^−1^ and achieved the lowest normalized activity value of RhB (0.50 mM·m^−2^). When urea was applied as an additive (BiOCl_U), similar results were obtained as for BiOCl_CTAC sample: the quality of the surface increased, because with smaller specific surface area (8.00 m^2^·g^−1^) degraded more RhB/surface units (3.08 mM·m^−2^). This could mean, that the cube-shaped hierarchical structures could be advantageous for the photocatalytic processes. Studying the BiOCl_TU sample a similar conclusion (to the case of BiOCl_PVP) was drawn: with higher specific surface area (25.00 m^2^·g^−1^), achieved lower RhB degradation/surface unit (1.20 mM·m^−2^).

The normalized degradation values of phenol were also evaluated. In case of BiOCl_SDS and BiOCl_PVP lower phenol degradation units were obtained (1.53 mM·m^−2^ and 1.60 mM·m^−2^). When BiOCl_ Ø, BiOCl_TU and BiOCl_CTAC was investigated near equal values were obtained reported to the surface, showing that apparently the quality of the materials’ surface remained unchanged using these additives. The highest phenol degradation/surface unit was obtained in case of the BiOCl_U. The information gained from the phenol normalized degradation values was that the degradation of phenol was more surface independent than that of RhB. This was expected as phenol is degraded in the aqueous media by OH radicals, while RhB contacts directly to the surface of the catalyst.

After the degradation of phenol by UV the catalyst was recovered, the color of the semiconductors changed from white to gray. Raman measurements was taken to study and to explain the reason of this phenomenon. In case of the BiOCl sample series, no correlation was found between the photocatalytic efficiency and (101), (110), and (102) crystallographic plane ratios.

The kinetics of the phenol degradation under UV light was investigated to see whether the reaction rate was responsible for the observed conversion values or other important aspects are playing an important role. The pseudo-first order rate constants listed in the manuscript (Table 1) showed that there was no specific correlation (ranging from 0.1603 to 0.2259 mol∙L^−1^∙min^−1^) among the values, as they are being evaluated for the first 5 data points at which the pseudo-first order reaction can be considered valid (later on the formation of the intermediates are modifying the order of the reaction). This means that the final degradation efficiency values are mostly dependent on the degradation of the intermediates rather than that of phenol. It is important to mention also that the activity of the catalysts was reproducible for at least four catalytic runs.

### 3.5. The Reason behind the Photocatalytic Activity of the Samples

#### 3.5.1. The Effect of the Surface Tension

The surface tension values were calculated for each synthesis solution to understand better the effect of additives on the overall properties of the materials. The solvation energy of precursors and solution in the surface layer could result the growth of the preferred crystal facet, so it is mandatory to investigate the role of the surface tension.

As it can be seen in Figure 5, when the surface tension of the synthesis liquid increases, the specific surface area of the final product increased as well, but the average hierarchical crystal size decreased at the same time. The reason for this can be linked with the formation and size of micelles, which may become smaller at higher surface tension values. Higher surface tension values resulted beneficial semiconductor properties, such as smaller primary crystallite size and higher specific surface area, which are also beneficial for the photocatalytic processes. The results are also confirming the correlation between the synthesis mixtures’ surface tension and the final photocatalytic activity under UV light (both RhB and phenol degradation), meaning that higher conversion values of RhB and phenol increased with the applied surface tension values. Similar trend was observed between the surface tension and the band gap values, namely that there is a linear dependence between them. 

It seems that the surface tension of the synthesis liquid tied together the UV-phenol photocatalytic activity with the hierarchical crystal size, band gap energy and specific surface area. The latter one could be explained a very known fact: the smaller the size of the hierarchical particle, the larger its specific surface area, which means that the accessibility of the catalytically active sites is enhanced, making possible more efficient photocatalytic reactions.

The band gap energy is also a crucial parameter as it seems that with the increase of this value the activity towards phenol degradation also increases most probably due to the lower recombination rates of the charge carriers. This is possible as it is known that as the band gap of the semiconductors are narrowing the recombination rate becomes preferred (the main reason behind the lower efficiency of visible light photocatalysts).

#### 3.5.2. Evaluation of the Raman spectra

Figure 6 shows the Raman spectra of BiOCl semiconductors before and after the degradation process. Before the degradation (left), a signal at 142 cm^−1^ was detected and assigned to the A_1g_ internal Bi-Cl stretching and another one at 197 cm^−1^ which originated from the E_g_ characteristic vibrations of internal Bi-Cl.

After the degradation of the phenol under UV light irradiation (right), new signals appeared (Bi-O at 159 cm^−1^ and 211 cm^−1^; and Bi-Bi at 109 cm^−1^), while in all the BiOCl samples, the Bi-Cl characteristic vibrations’ intensity decreased.

From the clear appearance of Bi-Bi vibrations it can be concluded that some oxygen atoms from the Bi-O-Bi bridges disappeared. On the other hand, no signal of metallic bismuth was obtained, making possible the scenario of oxygen vacancy appearance which also can cause the color change in the investigated semiconductors. 

If a Bi-O-Bi entity is present at the surface during a photocatalytic reaction the oxygen may leach from the structure creating reduced Bi centers ([Bi–*–Bi] clusters), while the oxidation of the pollutant occurs. The mentioned clusters are very unstable, and it could be oxidized back by the molecular O_2_ present in the aqueous media and therefore new Bi-O bonds can be formed, which are also shown in the Raman spectra. However, the reaction rate of the two reactions (creation of reduced centers vs. reoxidation) may differ in the favor of the reduction, and that is why the color change was still observable after the photocatalytic tests.

The reason of the disappearance of the Bi-Cl bonds’ specific signals could be the distorted crystal structure. When the oxygen departs from the crystal structure not only [Bi–*–Bi] clusters are formed, but also distorts the Bi-Cl bonds as well, while considering the fact that Raman gives more surface related information.

From another approach, the reason of the disappearance of the Bi-Cl bonds and the formation of Bi-O bonds could be the diffusion of oxygen into the crystal lattice and distortion of the structure, however, in this case a local oxidation must occur, or a local molecular O_2_ formation is mandatory. This phenomenon is called the surface dependent oxygen activation. Two types of reactive oxygen species have been identified so far: electrophilic entities, such as (superoxide (•O^2 −^) or peroxide (O_2_
^2−^) and entities (e.g., lattice-Oand terminal or bridging oxygen containing groups). Especially, electrophilic species are responsible for increased photocatalytic activity, while nucleophilic species for the catalytic selectivity. In redox reactions, the crucial steps are the transfer of the substrate oxygen atoms into the adsorbed reactants and the filling of the vacancies by dissociating O_2_ originated from the reaction media. This mechanism is known as the Mars–van Krevelen phenomena and it is known for several compounds such as for CeO_2_, V_2_O_3_, BiOCl and TiO_2_ [28,29,30,31].

#### 3.5.3. XPS Investigation

As the Bi4f XPS spectra of the samples were investigated, the following bismuth entities were identified: Bi^3+^ (159.3 eV), Bi^5+^ (160.1 eV) and Bi^0^ (157.7 eV), as shown in Figure 7. The appearance of the latter component was not surprising as the Raman spectra indicated a defective structure (more precisely an oxygen deficient crystal lattice, which manifests in Bi-Bi clusters, as shown and discussed above). However, it should be mentioned that all the samples showed the same structural features, namely Bi^3+^ (159.3 eV), Bi^5+^ (160.1 eV) and Bi^0^ (157.7 eV). Also, it was found interesting that BiOBr prepared by the same synthesis route showed mostly oxidized species (Bi^4+^ and Bi^5+^) as discussed in our previous paper, meaning that a similar synthesis route applied to halogenates may induce quite different property changes. The concentration of the species was (91.2–93.5% Bi^3+^, 3.4–3.7% Bi^0^ and 2.8–5.4% Bi^5+^) was quite low except for Bi^3+^, meaning that the behavior of the surface was independent and did not show any specific changes within the different BiOCl series. Th extremely low content of reduced Bi can be linked with the instable nature of these reduced sites, which was also observable by naked eye, as the samples regained their white color from greyish after shutting of the UV irradiation source. The O1s and Cl2p spectra of the samples were also considered but did not show any significant information, therefore will be not presented here.

## 4. Conclusions

Several BiOCl samples were obtained using different additives. From the investigations regarding the morphology, it was observed that the catalysts showed a hierarchical structur (microspheres/flowers and cubes), which were comprised of individual plates. The additives affected on the (102)/(110), (101)/(110) and (102)/(101) crystal facet ratios and the primary crystallite size. It was found that by increasing (102) crystallographic plane’s dominance resulted a redshift in the band gap energy of BiOCl materials, which implies, that the (102) crystallographic plane could have a lower band gap energy compared to (101). Despite of the previously mentioned link between crystal orientation and band-gap, the activity did not correlate with the orientation of the crystals. The surface tension of the synthesis liquid was accountable for the difference in morphological and structural peculiarities, including specific surface area, hierarchical particle size, band gap and UV light-driven catalytic performance. The colour of the BiOCl semiconductors changed after the photocatalytic degradation of phenol under UV light irradiation. This change was studied with XPS and Raman spectroscopy and it was concluded that oxygen vacancies or surface dependent oxygen activation. The best photocatalytic activity towards phenol (both under UV and visible light) was achieved, when the highest surface tension liquid was the matrix for the synthesis procedure, resulting higher band gap energy, higher specific surface area, lower hierarchical particle size. The RhB degradation efficiency was not directly linkable to any of the investigated parameters.

## Figures and Tables

**Figure 1 materials-14-02261-f001:**
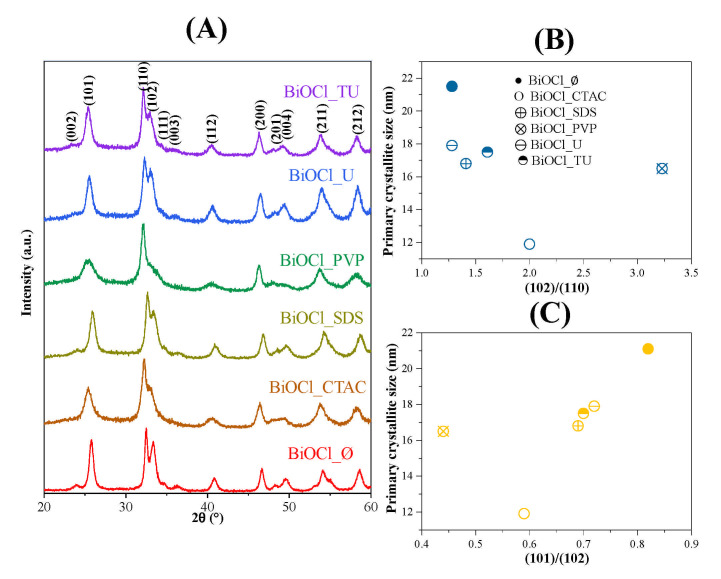
The XRD patterns of the BiOCl materials, using different additives (**A**) and the correlations between the primary crystallite size and (102)/(110) (**B**), (101)/(102) (**C**) and crystallographic planes’ ratio.

**Figure 2 materials-14-02261-f002:**
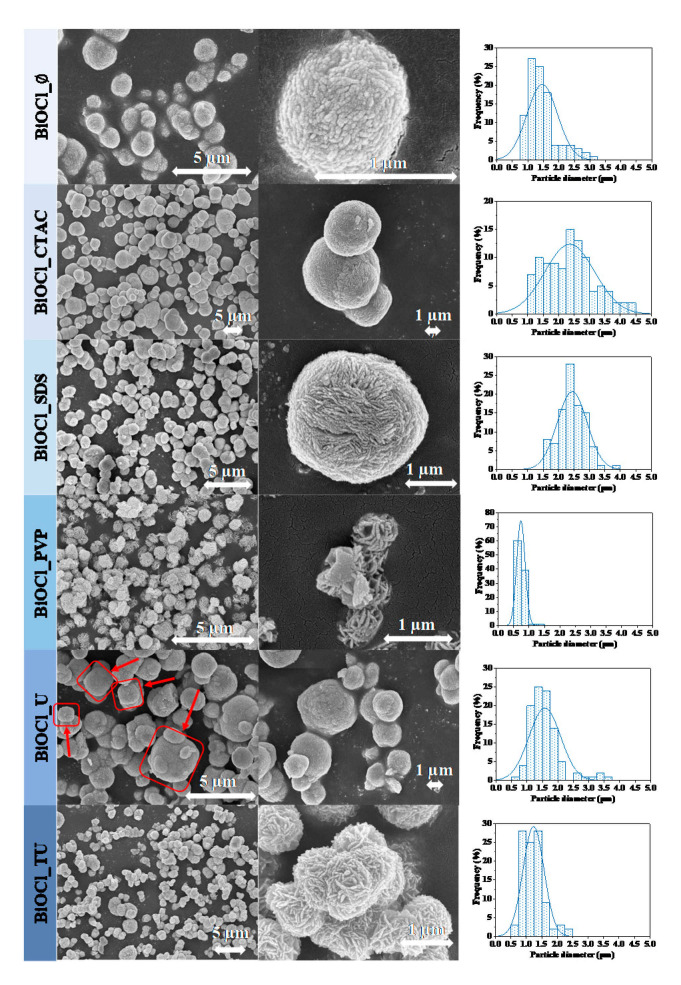
The SEM micrographs of the obtained BiOCl catalyst materials using different additives.

**Figure 3 materials-14-02261-f003:**
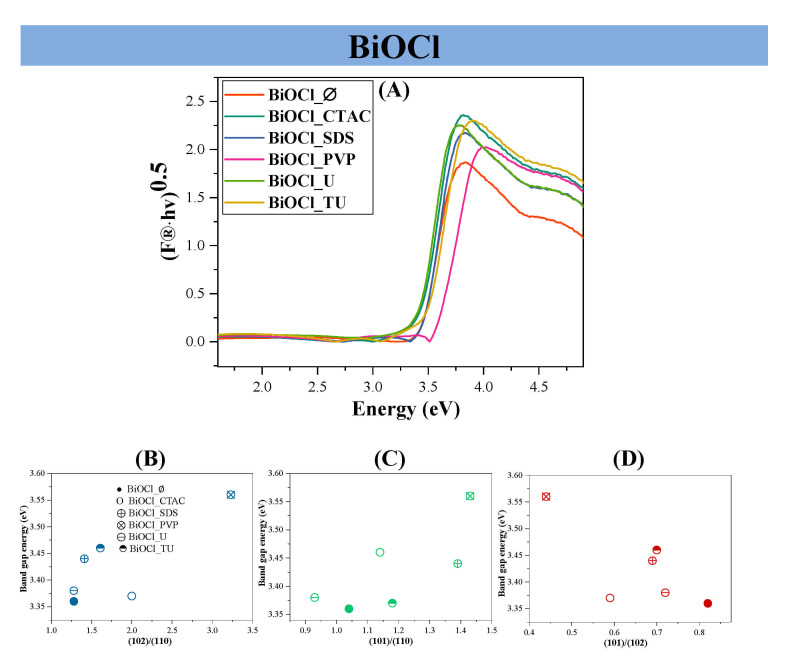
The diffuse reflectance spectra of the obtained BiOCl semiconductors (**A**); the correlation between the band gap values and (102)/(110) (**B**), (101)/(110) (**C**) and (101)/(102) (**D**) crystallographic plane ratios.

**Figure 4 materials-14-02261-f004:**
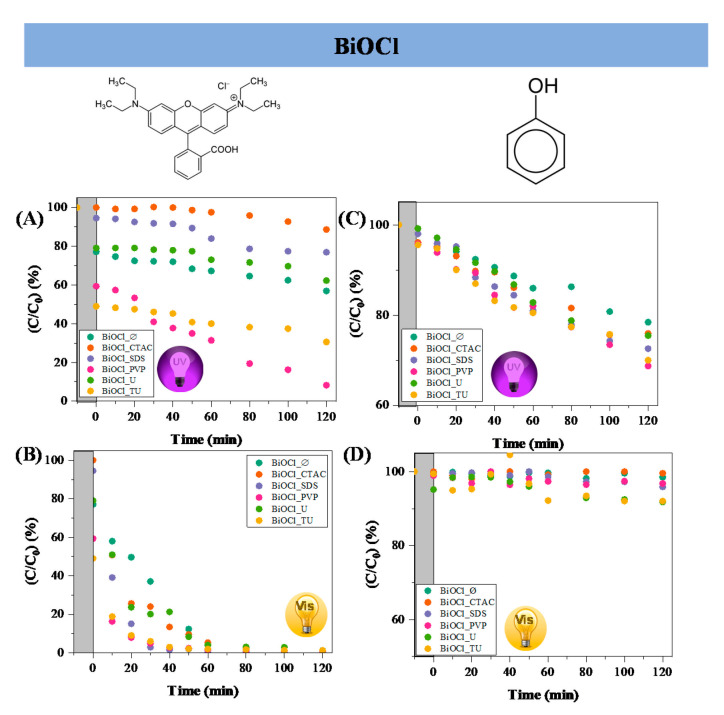
The photocatalytic degradation of RhB using BiOCl materials synthesized in the presence of different additives under UV-A (**A**) and visible light (**B**) irradiation; phenol photodegradation under UV-A (**C**) and visible light (**D**) with the same photocatalysts.

**Figure 5 materials-14-02261-f005:**
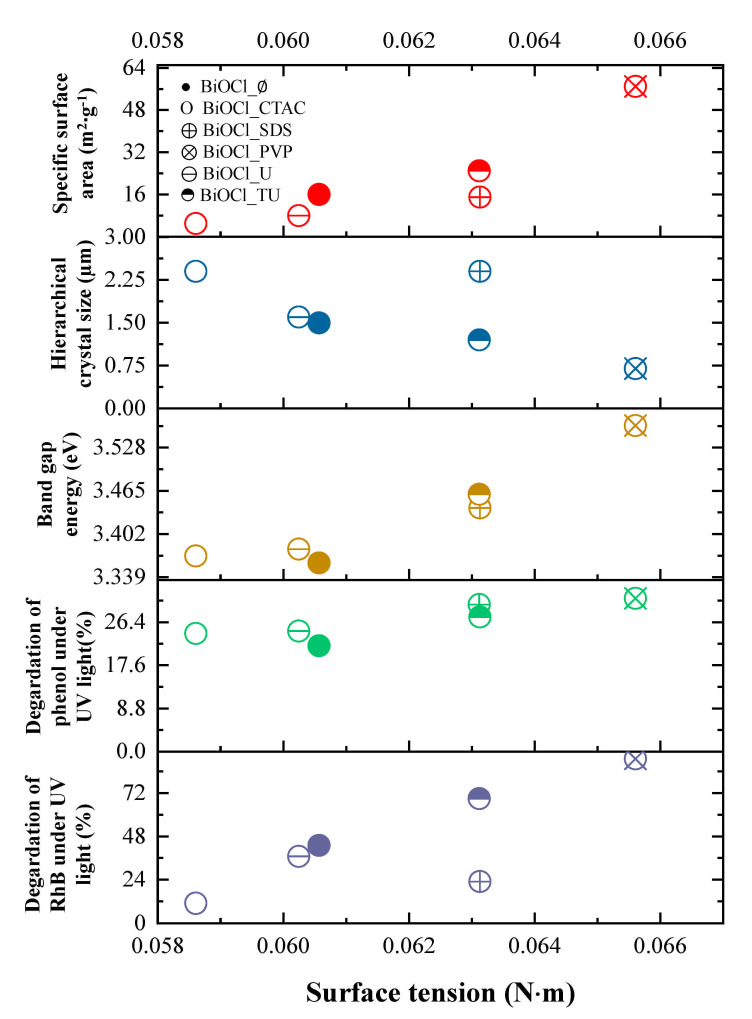
The influence of the surface tension on specific surface area (red), hierarchical crystal size (blue), band gap energy (brown), and the degradation of phenol (green) and RhB (purple) under UV irradiation.

**Figure 6 materials-14-02261-f006:**
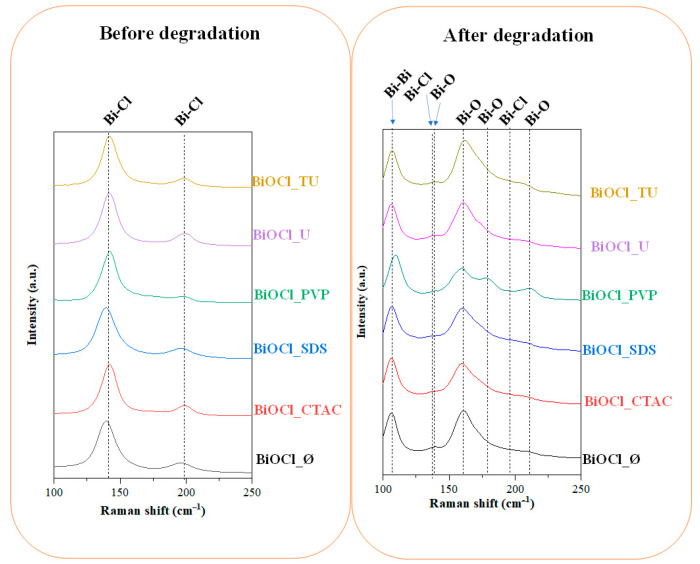
The Raman spectra of the obtained BiOCl semiconductors synthetized with different additives.

**Figure 7 materials-14-02261-f007:**
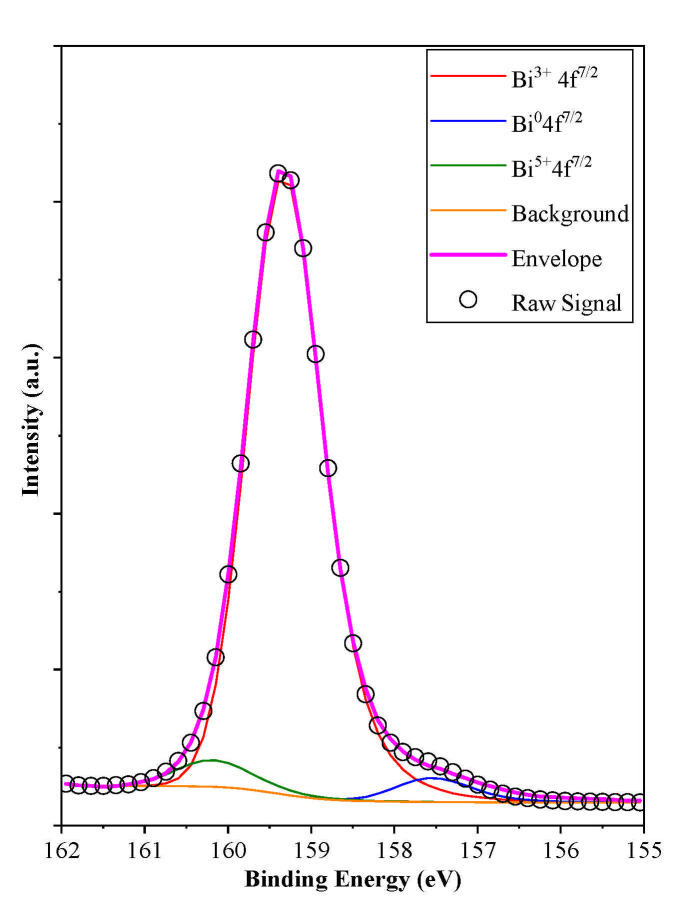
Bi4f ^7/2^ XPS spectrum of sample BiOCl_PVP, showing both reduced and oxidized bismuth species.

**Table 1 materials-14-02261-t001:** Morphological, structural, optical and photocatalytic properties, of the investigated BiOCl semiconductors.

Sample Name	Mean Primary Crystallite Size (nm)	Median Hierarchical Crystal Size (µm)	Band Gap Energy (eV)	(101), (102) and (110) Intensity Ratio.	Specific Surface Area (m^2^·g^−1^)	Conversion of Phenol (%) **		Conversion of RhB ** (%)		Adsorbed RhB (%)	Normalized Degraded Phenol (mM·m^−2^)	Normalized Degraded RhB (mM·m^−2^)	Kinetic Constant (mol L^−1^ min^−1^)
						UV	Vis	UV	Vis				
**BiOCl_∅**	21.1	1.5	3.36	0.82:1.00:0.78	16	21.6	1.5	43	-	22.7	1.35	2.69	0.1603
**BiOCl_CTAC**	11.9	2.4	3.37	0.59:1.00:0.5	5	24.1	0.0	11	99	0.0	4.82	2.20	0.1865
**BiOCl_SDS**	16.8	2.4	3.44	0.69:1.00:0.71	15	27.5	4.1	23	99	5.1	2.83	1.53	0.2259
**BiOCl_PVP**	16.5	0.7	3.56	0.44:1.00:0.31	57	31.3	3.2	91	99	40.4	0.55	1.60	0.2229
**BiOCl_U**	17.9	1.6	3.38	0.72:1.00:0.78	8	24.6	8.2	37	99	21.0	3.08	4.63	0.2178
**BiOCl_TU**	17.5	1.2	3.46	0.70:1.00:0.62	25	30.0	8.0	69	99	51.1	1.20	2.76	0.2063

** RhB showed an 8%/5%, while phenol 3%/1% of photolysis (UV/Visible).

## Data Availability

Data available on request due to restrictions. The data presented in this study are available on request from the corresponding author. The data are not publicly available due to the fact that it is stored on personal computers.

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
