# Peer review of "Hydrothermal Crystallization of Bismuth Oxychlorides (BiOCl) Using Different Shape Control Reagents"

_materials, 2021, doi:10.3390/ma14092261_

Round 1
Reviewer 1 Report
This work presents the hydrothermal crystallization of the bismuth oxychlorides (BiOCl) using different shape control reagents. The authors found a strong dependency between the surface tension of the synthesis solutions and the overall morphostructural parameters.This effect is investigated deeply and the conclusions are supported by the experimental data. However, some suggestions should be considered by the authors.
1) The authors must describe the abbreviations when used for the first time in the text. For example, on row 32 "BiOX".
2) The caption of Figure 1 is:
"The XRD patterns of the BiOCl materials, using different additives (A) and the correlations between the primary crystallite size and (102)/(110) (B), (101)/(110) (C) and (101)/(102) crystallographic planes’ ratio".
This is different from the figure content. In Figure 1, the (101)/(110) ratio is missing, and the (101)/(102) ratio is presented in (C).
3) The authors must give more information about the SEM images. They should try to adjust the sizes of the SEM images. To make the comparison possible, images have to be chosen with the same magnification and the scale has to be the original one. At the present form, all conclusions derived from these images are speculative.
4) In Figure 3, for (C), the values for the X axis have to be up to 3.60, the same as for (B) and (D). The authors must choose either a decimal point or a decimal comma.
Author Response
Most Respected Referee,
We are grateful for your efforts to provide us substantial comments and suggestions. Please consult the attached file for our responses for each of the raised issues, according which we have modified the manuscript. Sincerely yours,
The Authors

Reviewer 2 Report
The authors report an experimental study of the effect of various additives on the morphology and photochemical activity of BiOCl. The paper is difficult to read, partly because of poor English and partly because of the lack of logic.
The conclusions are not clear.
The main question is: is BiOCl suitable as a photochemical catalyst or does it decompose as quickly as the substance that is supposed to decompose?
Does the activity of BiOCl mainly depend on the structure of crystallites or their aggregates?
Minor:
- “These materials have tetragonal matlockite structure, where each bismuth atom is surrounded by four oxygen and four chloride atoms, building [Cl-Bi-O-Bi-Cl] layers held together by Van der Waals forces.” Show the crystallographic planes on a figure.
- “In contrast, when additives were used during the BiOCl synthesis, the intensity ratios of the (101), (110) and (102) crystallographic planes changed, specifically the reflections of (101) and (102) (Table 1S). These changes may arise from morphological differences, which would impact the photocatalytic activity.” What other reason could cause these changes?
- Correct the captions of Figures 3 and 4.
Author Response

(The authors gave the same response as above.)

Reviewer 3 Report
This paper discusses bismuth oxychloride samples synthesized using solvothermal crystallization and different additives. It focuses on comparison of the effects of the different additives on the surface tension, crystal structure, morphology, band gap energy, surface tension of the synthesis mixture and investigation of the photo-catalytic activity of the as-obtained semiconductors towards phenol and RhB (Rhodamine B).
Some shortcoming and missing of the paper are the following:
- Table 1S, pointed into text several times (see, for example, row 163), is absent in the paper.
- At ordinate axis of low figure in Fig. 5 must be “Degradation of RhB under UV light” (no phenol).
Author Response

(The authors gave the same response as above.)

Reviewer 4 Report
This is an interesting paper discussing the impact of synthesis additives on the characteristics of the final BiOCl. The compounds are well characterized. I do have some comments and suggestions given below.
A general comment: Please explain in Figure captions what is seen in the different figures a, b, c etc., as this would ease the reading of the figures.
Lines 91-92, how to understand the following "Then 0.42 g (Bi:X = 1:1) of KCl (VWR 99.0 %) was added."
Lines 95-96, is the materials washed with a mixture of water and ethanol or first water, then ethanol?
Considering the fixed wavelength during RhB degradation analysis, it would be useful to add a full UV-vis spectrum to prove degradation and not just modification of the RhB molecule as deethylation decreases the optimum wavelength. See e.g., DOI: https://doi.org/10.1021/jp905173e
I recommend to move the table from Supplementary Information to the actual paper as the information it makes it easier to compare the data from the various produced BiOCl catalysts, and the table is important due to the continous referring to this table.
In Section 3.1, regarding the changes in crystallite size and crystal planes, it would add great value to the paper if it was discussed why the different additives causes these changes - is there an effect of additive size, hydrophilicity or something else?
Line 226: Do you have any possible explanations why the BiOCl produced with CTAC did not follow the trend in Fig 3?
Line 236: It is stated that the catalytic activity decreases for BiOCl produced with additives, however, Table S1 shows a higher conversion? Please explain this.
Lines 241-243: Have you taken into account the adsorption of dye to the catalyst when calculating the RhB conversion? If 40-50 % is adsorbed, it cannot be 91 % conversion? I suggest to investigate whether RhB is adsorbed to the catalyst after photocatalytic experiments, or it actually is degraded. This can, e.g., be performed using FT-IR experiments as RhB has very characteristic peaks.
As the degradation is followed in Fig. 4, I suggest to add the kinetic constants for all experiments. These values could either be added in Table S1 or a new table could be provided.
Have you investigated as a baseline if phenol or RhB degrades in UV og vis light without adding the catalyst?
The conclusion should be more focused on what is shown to be better, e.g., which crystal shape improves the photocatalytic degradation.
Author Response

(The authors gave the same response as above.)

Round 2
Reviewer 2 Report
The main questions were not answered.
Is BiOCl suitable as a photochemical catalyst or does it decompose as quickly as the substance that is supposed to decompose?
Does the activity of BiOCl mainly depend on the structure of crystallites or their aggregates?
Author Response
Most respected referee,
Before starting to answer the issues raised by the referee, we deeply apologize for ignoring those two comments of the referee which were highlighted in the response process. The authors did not intentionally made this mistake as it is our goal to take part in an honest and critical review process to improve the quality of research papers. Therefore, with the deepest respect please allow us to respond appropriately.
Sincerely yours,
The authors

Reviewer 4 Report
The authors replied to all of my concerns in a satisfactory manner, and I recommend the paper to be accepted.
Author Response
Most respected Referee,
We are grateful for your positive answer and for your given critics and suggestions during the peer review process as it improved our manuscript. Best regards,
The authors